# Human Induced Pluripotent Stem-Cell-Derived Cardiomyocytes as Models for Genetic Cardiomyopathies

**DOI:** 10.3390/ijms20184381

**Published:** 2019-09-06

**Authors:** Andreas Brodehl, Hans Ebbinghaus, Marcus-André Deutsch, Jan Gummert, Anna Gärtner, Sandra Ratnavadivel, Hendrik Milting

**Affiliations:** 1Erich and Hanna Klessmann Institute, Heart and Diabetes Center NRW, University Hospital of the Ruhr-University Bochum, Georgstrasse 11, D-32545 Bad Oeynhausen, Germany (H.E.) (J.G.) (A.G.) (S.R.); 2Department of Thoracic and Cardiovascular Surgery, Heart and Diabetes Center NRW, University Hospital Ruhr-University Bochum, Georgstrasse 11, D-32545 Bad Oeynhausen, Germany

**Keywords:** induced pluripotent stem cells, cardiomyopathies, cardiovascular genetics, cardiomyocytes, ARVC, DCM, HCM, RCM, NCCM, LVNC

## Abstract

In the last few decades, many pathogenic or likely pathogenic genetic mutations in over hundred different genes have been described for non-ischemic, genetic cardiomyopathies. However, the functional knowledge about most of these mutations is still limited because the generation of adequate animal models is time-consuming and challenging. Therefore, human induced pluripotent stem cells (iPSCs) carrying specific cardiomyopathy-associated mutations are a promising alternative. Since the original discovery that pluripotency can be artificially induced by the expression of different transcription factors, various patient-specific-induced pluripotent stem cell lines have been generated to model non-ischemic, genetic cardiomyopathies in vitro. In this review, we describe the genetic landscape of non-ischemic, genetic cardiomyopathies and give an overview about different human iPSC lines, which have been developed for the disease modeling of inherited cardiomyopathies. We summarize different methods and protocols for the general differentiation of human iPSCs into cardiomyocytes. In addition, we describe methods and technologies to investigate functionally human iPSC-derived cardiomyocytes. Furthermore, we summarize novel genome editing approaches for the genetic manipulation of human iPSCs. This review provides an overview about the genetic landscape of inherited cardiomyopathies with a focus on iPSC technology, which might be of interest for clinicians and basic scientists interested in genetic cardiomyopathies.

## 1. Introduction

At the beginning of this century, the human genome project was finished [1]. The development of next generation sequencing (NGS) technologies significantly reduced the price and time, allowing for efficient genome and exome analyses, even in clinical routine procedures. However, even 20 years later, the clinical interpretation of genetic sequence variants (GSVs) is still challenging because the functional and structural impact of many variants is unknown. Therefore, multi-disciplinary approaches are often necessary for the interpretation and functional analysis of novel GSVs [2]. At present, in clinical routine procedures, the pathological impact of GSVs is classified due to standards and guidelines of the American College of Medical Genetics and Genomics (ACMG) [3].

Cardiomyopathies are diseases that affect the heart muscle, leading to functional and structural abnormalities [4], and are the main indication for heart transplantation (HTx) [5]. Beside environmental factors, like myocarditis or cardiotoxicity of cancer drugs, non-ischemic cardiomyopathies often have a genetic etiology with dominant inheritance. However, because pathogenic mutations in more than 100 different genes are associated with non-ischemic cardiomyopathies, the interpretation of novel GSVs is still challenging [6]. Moreover, little is currently known on digenic, or even polygenic, etiologies of cardiomyopathies [7]. Incomplete penetrance, different expressivity, and pleiotropy make the clinical interpretation even more challenging.

Functional analyses using adequate cell and animal models can lead to a more sophisticated interpretation of GSVs, which might be not only relevant for genetic counseling but also for the development of personalized therapies. According to the ACMG guidelines, in vitro or/and in vivo functional analyses provide strong criteria (PS3) for the classification of GSVs [3,8]. However, the generation of animal models is still time consuming and expensive. Moreover, in some cases, human cardiomyopathies cannot be modeled using animal models because of species differences. For example, *TMEM43*-p.S358L is a mutation with full penetrance in several families with arrhythmogenic cardiomyopathy (ACM) [9,10,11]. In contrast, the *Tmem43* knock-out, as well as the knock-in mice carrying this specific mutation, do not develop an ACM phenotype [12]. Because of these limitations, human iPSC-derived cardiomyocytes are unprecedented research tools to model and investigate genetic cardiomyopathies.

Here, we provide an overview about the genetic landscape of inherited cardiomyopathies and summarize the development of important human iPSC lines for modelling human cardiomyopathies in vitro. In addition, we review the differentiation into cardiomyocytes and discuss relevant methods used for the cellular and molecular characterization of human iPSC-derived cardiomyocytes.

## 2. Clinical Background

In clinical cardiology, cardiomyopathies are classified into five major structural subtypes (Figure 1). Dilated cardiomyopathy (DCM, MIM #604145) is mainly characterized by left-ventricular dilation in combination with a decrease of the wall diameter [13]. These structural changes decrease the cardiac ejection fraction. Hypertrophic cardiomyopathy (HCM, MIM #160760) is characterized by the hypertrophy of the ventricular walls and/or the septum [14], leading to a reduced cardiac output. Restrictive cardiomyopathy (RCM, MIM #115210) is caused by an increase in ventricular stiffness, leading to dilated atria and diastolic dysfunction [15]. Hyper-trabeculation of the left ventricular wall is a hallmark for (left-ventricular) non-compaction cardiomyopathy (NCCM, MIM #604169) [16]. It mainly affects the left ventricle, but isolated right ventricular or biventricular forms of NCCM have been reported [17]. Ventricular arrhythmias and predominant right or biventricular dilation are the main clinical symptoms of ACM (MIM #609040) [18]. The fibro fatty replacement of the myocardial tissue is a pathognomonic feature characteristic of ACM [19]. However, at the early stage of the disease, structural changes may be absent or subtle [20]. Because ACM is a progressive disease, left ventricular involvement develops frequently at a later stage [21].

## 3. Genetic Basis of Inherited Cardiomyopathies

Thirty years ago, Seidmans’ group discovered the first pathogenic mutation in *MYH7,* encoding for β-myosin heavy chain, in a four-generation family, in which several members developed HCM [22]. At present, genetic variants have been described in more than 100 different genes associated with non-ischemic cardiomyopathies or syndromes with cardiac involvement such as Marfan or Leopard syndrome (for an overview, see Table 1). Of note, the spectrum of affected genes and mutations partially overlaps between the different non-ischemic cardiomyopathies (Figure 1). For example, mutations in *DES*, encoding the muscle specific intermediate filament protein desmin, might cause DCM [23,24], HCM [25], ACM [26,27], RCM [28], or NCCM [29,30,31]. Similarly, mutations in *TTN*, encoding the giant sarcomere protein titin, can also cause different types of structural, non-ischemic cardiomyopathies [32,33,34]. However, the molecular reasons why mutations in the same gene can cause different cardiac phenotypes are largely unknown.

From a genetic point of view, non-ischemic cardiomyopathies are quite heterogeneous [35,36,37]. However, the different non-ischemic cardiomyopathies are characterized by an accumulation of mutations in a distinct set of genes encoding for proteins that are essential for cardiomyocyte function. For example, HCM is mainly caused by mutations in genes encoding sarcomeric proteins such as *MYH7* or *MYBPC3* (Figure 1). Further mutations in other genes, encoding sarcomere proteins, like *TPM1* [38], *TNNC1* [39], *TNNI3* [40], *TNNT2* [38], *FHL1* [41,42], or *ACTC1* [43], have also been identified in patients with HCM (Table 1). In addition, in rare cases, mutations in genes encoding for Z-disc proteins, like *ACTN2* [44] or *FLNC* [45], or genes encoding for proteins involved in the Ca^2+^-homeostasis like *PLN* [46], are also known to cause HCM (see Figure 1).

*TTN* is the most prevalent DCM-related gene with truncating *TTN* mutations identified in about 20–25% of DCM patients [32,47]. However, several other genes with a lower prevalence can also cause DCM. Besides, mutations have been identified in genes coding proteins of the sarcomere (e.g., *MYH7* [48]), the cytoskeleton (e.g., *DES* [23,24]), the nuclear lamina (e.g., *LMNA* [49]), ion channels (e.g., *SCN5A* [50]), and transcription (e.g., *EYA4* [51]) or splicing factors (e.g., *RBM20* [52]) (Table 1). *RBM20* mutations cause an aggressive early onset phenotype including arrhythmias, sudden cardiac death, and DCM, especially in males [53]. In total, mutations associated with DCM have been described in about 80 different genes (see Figure 1 and Table 1).

NCCM is the third most frequent non-ischemic cardiomyopathy [54,55] and can occur as a primary cardiomyopathy or can be part of a syndromic disease like the Barth syndrome (MIM, #302060) [56]. Mutations in over 20 different genes having a significant overlap with HCM- or DCM-associated genes have been described in NCCM patients so far (see Figure 1 and Table 1). Comparable to HCM, the most prevalent NCCM-associated genes are *MYH7* and *MYBPC3* [57], which encode sarcomeric proteins (Table 1).

ACM is mainly caused by mutations in genes, encoding structural components of the cardiac desmosomes, and adherens junctions [26,58,59]. The cardiac desmosomes are cell–cell junctions mediating the adhesion of the cardiomyocytes [60]. In about 50% of the ACM patients, one or more mutations in desmosomal genes can be identified [26,59,61] (Table 1). Cardiac desmosomes are linked through the intermediate filaments formed mainly by desmin (*DES*) with several other cell organelles like the Z-bands or the nuclei. Of note, mutations in the *DES* gene can also cause ACM by abnormal cytoplasmic desmin aggregation [26,62]. In addition, mutations in genes of the nuclear envelope like *LMNA* [63], *TMEM43* [9,10], or *LEMD2* [64] are associated with ACM (Table 1). Furthermore, some rare mutations in non-desmosomal and non-nuclear genes like *RYR2* [65,66], *PLN* [67], or *ILK* [68] have been identified in ACM patients.

Currently, the genetic etiology of RCM is poorly characterized. Recently, Kostareva et al. and Gallego-Delgado et al. genotyped two small cohorts of unrelated RCM index patients and identified likely pathogenic or pathogenic mutations in 50–75% of them [69,70]. The majority of affected RCM genes, which partially overlap with the group of HCM-associated genes, encode for sarcomere or cytoskeleton proteins (see Figure 1 and Table 1). The first RCM-associated mutation was identified in *TNNI3*, encoding cardiac troponin I [71]. More recently, there is growing evidence that *FLNC* mutations, encoding the cytolinker protein filamin-C, are frequently associated with RCM [72,73,74,75,76].

In summary, a relevant amount of all non-ischemic cardiomyopathies have a genetic etiology. Although in most cases, cardiomyopathies are inherited monogenetically, the underlying genetic landscape is complex, diverse, and currently only partially known.

## 4. Generation of Patient-Specific-Induced Pluripotent Stem Cells Via Reprogramming

In the 1960s, Gurdon et al. cloned *Xenopus laevis* for the first time [274,275]. Consequently, Gurdon was awarded the Nobel Prize in medicine in 2012, together with Yamanaka [276]. The cloning of mammals by nuclear transfer from somatic cells into enucleated unfertilized mammalian eggs over twenty years ago demonstrated that the cellular differentiation can be artificially turned back into a pluripotent state [277]. The next breakthrough was the identification of essential reprogramming factors by the Yamanaka group [278,279]. Initially, reprogramming was performed with 24 candidate transcription factor genes. Out of these, four critical genes were identified to be crucial for iPSC generation: *Sox2*, *Oct4*, *Klf4*, and *c-Myc* [278]. Depending on the donor cell type, the set of reprogramming factors can vary since specific cell types might endogenously express some of the necessary factors. For example, *c-Myc* is not required for the reprogramming of fibroblasts [280].

Different delivery methods were developed for reprogramming of somatic cell types like fibroblasts, lymphocytes, keratinocytes, urine-derived, or intestinal cells into iPSCs (see Figure 2). Initially, iPSCs were generated using retroviral transduction [278,279,281]. The Moloney-based retroviral vector system used by the Yamanaka lab has the advantage of undergoing silencing in the iPSCs state but is restricted to dividing cell types. Therefore, lentiviruses were used to improve the transduction efficiency of dividing and non-dividing cell types. However, after lentiviral transduction, the expression of the reprogramming factors are poorly silenced [282,283], leading to difficult differentiation of these iPSCs [284]. Therefore, inducible systems were used, allowing for the silencing of the Yamanaka factors in iPSCs [284,285].

However, usage of integrating viral systems enhances the risk for insertional mutagenesis, limiting their application [286]. Furthermore, the transgene reactivation of c-Myc showed increased tumorigenicity in chimeric mice [280], limiting the usage of iPSCs for clinical approaches. To overcome these limitations, non-integrating delivery methods have been developed. Transient transfection of the PiggyBac transposon with a Cre-mediated excisable system was one of the first non-integrating methods (Figure 2). Minimized genome modification, in combination with silencing of the reprogramming factor expression in the iPSC state, are the main advantages of this system [287]. Another approach is the adenoviral transduction leading to an overexpression of the reprogramming factors in the host cells without genomic integration [288]. Transient transfection or electroporation with episomal plasmids encoding the reprogramming factors is an alternative method to produce virus-free iPSCs [289] (Figure 2). However, the efficiency of this delivery method is quite low [290]. More promising non-inserting delivery methods include the use of Sendai viruses [291], which are RNA viruses that do not enter the nucleus, thereby decreasing the risk of genomic insertion.

Reprogramming using miRNAs that are specifically expressed in embryonic pluripotent stem cells (ESCs) can enhance the reprogramming efficiency [292]. For example, the miR302/367 cluster is highly expressed in pluripotent cells, but not in differentiated cells, and its promoter is transcriptionally regulated by the reprogramming factors Oct4 and Sox2 [293]. This cluster is functionally involved in regulation of the cell cycle and maintenance of pluripotency. Overexpression of the miRNA cluster miR302/367 can promote the reprogramming of somatic cells [294]. In combination with the reprogramming factors, a higher efficiency can be achieved [292]. Although RNA-based reprogramming methods show higher efficiency compared to Sendai virus and episomal methods, the reliability is significantly lower [295]. Non-integrating delivery methods provide iPSCs that are more applicable for clinical disease modeling. Besides the integrating and non-integrating delivery systems, DNA-free approaches with transgene free reprogramming have been established. Small compounds or recombinant reprogramming factors were used (Figure 2) [296,297]. For example, the histone deacetylase inhibitor valproic acid improves the reprogramming efficiency [298,299]. The efficient synthesis of large amounts of purified native recombinant proteins and the permeabilization of the plasma membranes are crucial for this reprogramming method [300]. More recently, the CRISPR-dCas9-based synergistic activation mediator (SAM) system has been developed and applied for reprogramming [301,302]. This system is based on a fusion protein of the enzymatic inactive form of Cas9 (dCas9) and a transcription activator domain forming an artificial transcription factor which, in combination with specific guide RNAs, is able to activate the transcription of endogenous genes with minimal off-target activity. Weltner et al. successfully used this system for the expression of different reprogramming factors to generate iPSCs [302].

In summary, different integrating and non-integrating approaches have been developed for reprogramming different cell types into iPSCs to improve the efficiency and to reduce the risk of further genomic alterations (see Figure 2).

## 5. Genetic Modification of Induced Pluripotent Stem Cell Lines

Besides the generation of human iPSCs from the primary cells of mutation carriers by direct reprogramming [278,281], specific genetic mutations can also be inserted using genome editing techniques like clustered regularly interspaced short palindromic repeats associated protein 9 (CRISPR-Cas9) [303], CPF1 [304], or transcription activator-like effector nucleases (TALENs) [305,306]. In addition to genome editing approaches, iPSCs or the differentiated cardiomyocytes can be genetically modified by overexpressing specific mutant proteins [307,308] or by decreasing the expression of specific mutant proteins, e.g., by RNA interference [309].

Using patient-derived iPSCs, it is sometimes challenging to correlate directly functional effects in vitro with the specific genetic variants because the genetic and epigenetic background of the cells is largely unknown [310]. In contrast to patient-derived iPSCs, which carry the sum of all genetic sequence variants of the affected patients, genome edited iPSC lines carry specifically inserted mutations. Therefore, the effects of particular mutations can be directly compared with their corresponding isogenic wild-type controls in genome-edited iPSCs.

Genome editing techniques like CRISPR-Cas9 are based on endonuclease activity, which insert double-strand breaks (DSBs) into the DNA double helix at specific sites. Different endogenous cellular repair mechanisms like non-homologous end joining (NHEJ) or homology directed repair (HDR) are used for the repair of these DSBs. However, NHEJ is an imprecise process, which might lead to the insertion, deletion, or substitution of nucleotides [311]. Indel variants frequently cause frameshifts, and consequently, premature termination codons (PTCs). PTCs are recognized by nonsense mediated RNA decay (NMD) degrading the mutant mRNA. Therefore, DSBs can be efficiently used to generate knock-out models. In contrast, HDR uses DNA template molecules for the specific repair of the DSBs. In combination with suitable donor molecules, e.g., single-stranded oligonucleotides or double-stranded DNA templates like PCR products or plasmids, HDR can be used to insert specific point mutations [312], small peptide-encoding tags [313], or even larger fluorescence proteins at specific positions [314,315,316]. Unfortunately, the ratio of HDR to NHEJ is low, limiting the efficiency of knock-in strategies [317]. Therefore, different approaches for inhibiting NHEJ or promoting HDR have been developed (for reviews, see References [317,318,319]). The delivery of donor template molecules in close proximity to the DSBs by coupling Cas9 with the donor molecule might be a promising strategy [320,321,322]. An alternative are dCas9-related base pair editors [323,324,325], which can be used to exchange relevant nucleotides at specific positions.

## 6. Differentiation of Human Induced Pluripotent Stem Cells into Cardiomyocytes

The human adult heart is a post-mitotic organ with a very limited capacity for regeneration [326]. Beside the murine, atrial cardiomyocytes-related HL-1 cell line [327], no further contracting human cardiomyocytes cell lines are therefore currently available. Because of ethical and technical issues, the isolation of primary human cardiomyocytes from human surgical material and their long-time culture is in most cases impossible. Primary cardiomyocytes isolated from rodent hearts have characteristic differences like a different electrophysiology in comparison to the human ones. Therefore, cardiomyocytes derived from human ESCs or iPSCs are the predominant human cell resource [328,329].

Originally, Zhang et al. described the differentiation of cardiomyocytes from human iPSCs [330]. Comparable to ESCs, human iPSCs form embryonic bodies in suspension that can be further differentiated into cardiomyocytes [330,331,332,333,334]. However, the efficiency of this process was limited. In addition, monolayers of iPSC-derived cardiomyocytes can be generated [335,336]. In vivo, cardiogenesis is a complex cellular and molecular process where different transcription factors, growth factors, and miRNAs are time dependently expressed and regulated [337,338,339,340,341]. Driven by discoveries from development biology, it has been recognized that different recombinant growth factors, e.g., BMP4, can also be used to increase the efficiency of in vitro differentiation into cardiomyocytes [342,343,344]. In addition, modulation of the *Wnt* pathway by small molecules, e.g., CHIR99021 and IWP2, efficiently increases the differentiation into cardiomyocytes about 90% [344,345]. Furthermore, metabolic selection by glucose depletion, in combination with lactate supplementation, can be applied for further accumulation of cardiomyocytes [346,347]. Recently, Zhao et al. developed a method for the differentiation and generation of heteropolar cardiac tissue with atrial and ventricular ends [348]. Talkhabi et al. has previously reviewed the differentiation of iPSCs into cardiomyocytes in detail [349].

## 7. Methods for the Functional Analysis of Cardiomyocytes Derived from Induced Pluripotent Stem Cells

Besides general histochemical or molecular methods, e.g., RNA-Seq or proteomics, specific techniques for the functional in vitro analysis of the electrophysiological and contractile properties of iPSC-derived cardiomyocytes are frequently used. Patch clamping and multiple electron arrays (MEAs) are frequently used for the electrophysiological analysis of iPSC-derived cardiomyocyte monolayers [350,351]. The application of Ca^2+^ specific fluorescence dyes, e.g., Indo1 or Fura-2, allows for the microscopic analysis of Ca^2+^ transients [352,353,354]. Additionally, voltage-sensitive fluorescence dyes like di-4-ANEPPS can be used for the analysis of the electrophysiological properties [355]. For the analysis of the contractile properties of iPSC-derived cardiomyocytes, microscopic techniques like traction force measurements have also been used [356]. Atomic force microscopy can also be applied for measuring the contraction forces of iPSC-derived cardiomyocytes [357,358]. Feaster and coworkers developed a method to culture iPSC-derived cardiomyocytes on Matrigel mattresses, allowing for the contractility measurement by cell shortening [359].

## 8. Overview about Existing iPSC Lines Carrying Cardiomyopathy Associated Mutations

In 2010, Carvajal-Vergara and co-workers published a landmark paper about the generation of an iPSC line carrying the heterozygous mutation *PTPN11*-p.T468M [360]. Mutations in *PTPN11* cause the Leopard syndrome [361,362], which is frequently associated with severe HCM [363]. Interestingly, these iPSC-derived cardiomyocytes were larger and presented an abnormal, nuclear localization of NFATc4 [360]. Members of the NFAT family are involved in the calcineurin-NFAT signaling regulating hypertrophy [364]. Since this original report, about 70 different iPSC lines carrying cardiomyopathy-associated mutations in several different genes have been generated (Table 2). The majority of these mutant iPSC lines have been used for phenotypic modeling of genetic cardiomyopathies using electrophysiological and/or contraction measurements (Table 2). Besides modeling genetic cardiomyopathies, iPSC-derived cardiomyocytes were also used for the modeling of non-genetic causes of cardiomyopathies, e.g., doxorubicin cardiotoxicity [263,365], hypoxia [366], peripartum [367], or diabetic cardiomyopathy [368,369,370,371], or even infection with *Trypanosoma cruzi* [372] or with coxsackievirus B3 [373].

In the beginning, iPSC lines generated from healthy probands were frequently used as controls for experiments. However, because different iPSC lines have a variable genetic background, this approach has limitations. Since the development of efficient genome editing technologies like CRISPR-Cas9 or TALENs [303], it is common to generate isogenic control lines [374]. Interestingly, the reverse approach by inserting specific mutations in iPSCs from healthy control persons is also sometimes used [375]. In some cases, the rationale of these studies is the functional characterization of specific cardiomyopathy-associated mutations, which might contribute to a pathogenicity classification. In addition, iPSC-derived cardiomyocytes were used for the development of therapeutic strategies, e.g., genome editing. An interesting application of iPSC-derived cardiomyocytes is the testing of specific gene therapeutic concepts [376]. For example, Gramlich et al. applied antisense-mediated exon skipping in iPSC-derived cardiomyocytes with a truncating *TTN* (*TTNtv*) mutation for restoring the expression of titin [377]. However, at present, it appears that some of the *TTNtv* do not lead to premature translation termination in failing human hearts [378]. Thus, iPSCs might therefore be useful in future to check and modulate possible read-throughs of *TTNtv* mutations as well. Similarly, Kyrychenko et al. used CRISPR-Cas9 to delete whole exons within the *DMD* gene to correct the reading frame [379]. Of note, this strategy restores contractility in the iPSC-derived cardiomyocytes [379]. Hopefully, the combination of iPSC-derived cardiomyocytes with adequate modern genetic engineering tools will contribute in future to the development of therapeutic options in the context of personalized medicine.

## 9. Limitations of Human Induced Pluripotent Stem-Cell-Derived Cardiomyocytes

Besides cardiomyocytes, the human adult heart consists of several different cell types like fibroblasts, endothelial cells, leukocytes, pericytes, and smooth muscle cells. It has been estimated that the proportion of cardiomyocytes in myocardial tissue is around 25–35%, indicating that the majority of the cardiac cells are non-cardiomyocytes [460]. However, the molecular and cellular interactions and interferences between the different cardiac cell types are poorly understood. In particular, under pathological conditions like inflammation or fibrosis, the cellular composition of the heart of cardiomyopathy patients can vary and might change over time. Therefore, it is in general challenging to model the complex cellular and molecular networks using iPSC-derived cardiomyocytes in vitro, although the artificial generation of cardiac tissue has been impressively improved during the last few years [461,462,463,464,465]. Besides these general limitations, iPSCs and iPSC-derived cardiomyocytes have some specific limitations, which are outlined in the following paragraphs.

### 9.1. Genomic Instability

Genomic instability of iPSCs can be a fundamental problem limiting the clinical application of iPSC-derived cells because of safety concerns [466]. Mayshar et al. showed that a significant portion of iPSC and ESC lines carry full or partial chromosomal aberrations [467]. However, even for in vitro analysis, genomic instability could be an important issue, especially in the context of modeling genetic diseases like cardiomyopathies. Therefore, novel iPSC lines should be genetically characterized in general. Karyotype analysis using Giemsa staining or comparative genomic hybridization arrays can be used to detect larger chromosomal abnormalities, while next generation sequencing assays can be applied for genetic analysis at the single nucleotide level.

Three different mechanisms contribute to the mutagenesis in iPSCs: besides the existence of genetic variants in the parental somatic donor cells, mutations can be introduced during reprogramming procedure or during the long-time culture of iPSCs [468]. Of note, mutations might accumulate in iPSCs over the culturing time [469]. Therefore, it is advisable to use early passages and to repeat analyses for genetic stability from time to time.

### 9.2. Heterogeneity of iPSC-Derived Cardiomyocytes

Although cardiac differentiation protocols for iPSCs have been improved significantly over recent years [345,470], it should be kept in mind that iPSC-derived cardiomyocytes are still a heterogeneous cell population. Especially for bulk down-stream applications like proteomics, genomics, or metabolomics, this might have a significant impact.

### 9.3. Cellular, Molecular, and Functional Differences of Adult Ventricular Cardiomyocytes and iPSC-Derived Cardiomyocytes

Even though human iPSC-derived cardiomyocytes are contractile cell types, there are important cellular, molecular, and functional differences compared to adult cardiomyocytes. The most obvious differences are the size and shape of iPSC-derived cardiomyocytes. Adult ventricular cardiomyocytes have a typical rod-like shape and are relatively large cells with lengths of about 100 µm and diameters of 10–25 µm [471]. In contrast, iPSC-derived cardiomyocytes are much smaller [472] and are morphologically heterogeneous. The geometry of iPSC-derived cardiomyocytes ranges from round to rectangular or polygonal shapes [473,474]. In adult ventricular cardiomyocytes, the sarcomeric structure is highly organized and the Z-bands are in parallel with the intercalated disc. On the contrary, iPSC-derived cardiomyocytes have a more irregular and amorphous sarcomeric organization with diverse orientations [462,475]. In human myocardial tissue, the closed-ends of the plasma membranes connect the cardiomyocytes longitudinally and these ends of the cardiomyocytes “cylinders” are called intercalated discs. Multi-protein complexes mediate the cell–cell interactions at the intercalated discs and are subdivided into desmosomes, adherens, and gap junctions [476]. Although desmosomes and adherens junctions are also formed in iPSC-derived cardiomyocytes [472,477], the cellular distribution of these cell–cell junctions are not conserved [478,479]. Another important difference is the number of nuclei. Whereas a significant number of the human cardiomyocytes in vivo are binuclear cells [480], iPSC-derived cardiomyocytes are mononuclear cells [481]. In addition, there are significant differences in contraction and electrical properties of iPSC-derived cardiomyocytes in comparison to adult ones [474]. In summary, the structural and functional properties of iPSC-derived cardiomyocytes are more similar to fetal cardiomyocytes than to adult cardiomyocytes [482]. To overcome these limitations, different natural engineering approaches were established to drive cardiomyocytes maturation. One method is to stimulate the cardiomyocytes with electrical or mechanical impulses [483]. The composition of the extracellular matrix can also affect the interaction of the CMs, therefore influencing the cellular behavior [484,485]. Another promising approach is the co-culture of iPSC-derived cardiomyocytes with non-cardiomyocytes, enabling a more likely cardiac environment with different cellular interactions [486]. Physical, chemical, electrical, and genetic factors are being tested as stimuli for further maturation [487]. However, maturation of iPSC-derived cardiomyocytes is incompletely understood at the molecular level and more studies are needed in future.

## 10. Testing of Gene Therapies Using iPSC-Derived Cardiomyocytes as in Vitro Models

An interesting research topic is the development of personalized therapeutic strategies for genetic cardiomyopathies *in vitro*. Beyond the opportunities that reprogramming technologies offer for therapeutic myocardial regeneration, iPSC-derived cardiomyocytes are a promising platform to develop and test different gene therapies for genetic, non-ischemic cardiomyopathies. In general, the pathomechanisms of inherited cardiomyopathies can be classified into loss of function (LOF) or gain of function (GOF) mechanisms. LOF can be caused by (haplo)insufficiency or by the expression of non-functional proteins. For example, several HCM-associated *MYBPC3* mutations cause haploinsufficiency [415,488]. GOF is caused by mutant and toxic proteins such as those shown for several *DES* missense mutations [489,490].

Genome editing using CRISPR-Cas9 or TALENs has been applied to repair different mutations in iPSC-derived cardiomyocytes. After the insertion of DSBs, iPSCs repair these DSBs using NHEJ or HDR. Template molecules like oligonucleotides, plasmids, PCR products, or even the second chromosome might be used for HDR. Recently, Ma et al. even applied CRISPR-Cas9 for the repair of a pathogenic *MYBPC3* mutation in human pre-implanted embryos [491]. However, because the efficiency of HDR is low, the direct repair of mutations in iPSCs via genome editing is challenging. Therefore, single iPSC clones were frequently generated in vitro and the direct translational transfer of this method is consequently limited. A second therapeutic strategy is exon skipping [492]. Exon skipping corrects the open reading frame (ORF) of an affected gene via skipping of the mutant or multiple exons and restores the expression of the truncated, but still functional, protein. For this approach, specific antisense oligonucleotides binding to the mutant exons can be used [493]. Besides its application in iPSC-derived cardiomyocytes carrying mutations in *DMD* [494] or *TTN* [377], antisense-mediated exon skipping was also directly applied in human patients with Duchenne’s muscular dystrophy [495]. Recently, Eric Olson’s group applied CRISPR-CPF1 or -Cas9-mediated genome editing for exon skipping in iPSC-derived cardiomyocytes [379,496,497]. Prondzynski et al. applied trans-splicing and total gene replacement for the artificial increased expression of *MYBPC3* in iPSC-derived cardiomyocytes carrying a heterozygous frameshift mutation in *MYBPC3* [419]. The authors used adeno-associated viruses (serotype 2/9, AAV2/9) for the transduction of iPSC-derived cardiomyocytes with 5′- and 3′-pre-trans-splicing molecules and the total cDNA of *MYBPC3*. However, the efficiency of the trans-splicing approach was low. In contrast, the total gene replacement strategy increased the *MYBPC3* expression to over 80% in comparison with wild-type controls and was able to prevent cellular hypertrophy [419].

The combination of the iPSC-derived cardiomyocytes platform with gene therapy tools is a promising therapeutic approach enabling pre-clinical demonstration of proof-of-principle for inherited cardiomyopathies.

## 11. Summary

Human iPSC-derived cardiomyocytes represent the only available human cellular model for the direct functional analysis of specific genetic cardiomyopathies and might therefore overcome the limitation of species differences. Impressive progress in the reprogramming and differentiation procedure during the last decade allows, in combination with novel genome editing techniques like CRISPR-Cas9, for the development of defined/patient specific cardiomyocyte models including generation of their isogenic control lines. In summary, iPSC-derived cardiomyocytes have been used for: (a) the characterization of genetic variants of unknown significance, which might be helpful for genetic counseling [375]; (b) analyses of the molecular pathomechanisms [415]; and (c) the development of specific therapies [377,497].

However, the cellular and molecular crosstalk between inflammatory cells, fibroblasts, myoblasts, and cardiomyocytes is difficult to model using iPSC-derived cardiomyocytes. Therefore, in our opinion, iPSC-derived cardiomyocytes should also be combined with animal models or with ex vivo investigations of explanted human myocardial tissue whenever possible to overcome the specific limitations of iPSC-derived cardiomyocytes.

Interestingly, for some genes like *DMD*, *PKP2*, *MYBPC3*, or *MYH7*, several different iPSC lines have been generated. In contrast, for rare cardiomyopathy genes, e.g., *TMEM43*, no iPSC lines have been developed yet. The genetic analysis in the past few decades has revealed a high heterogeneity of inherited, non-ischemic cardiomyopathies. In our view, it is therefore important to generate further novel iPSC lines also carrying mutations in rare cardiomyopathy genes to compare the molecular differences and commonalities leading to non-ischemic cardiomyopathies. Hopefully, iPSC-derived cardiomyocytes will contribute to unravelling the pathomechanisms of genetic cardiomyopathies and will help in efficient drug development in future.

Gene names follow the official guidelines of the HUGO Gene Nomenclature Committee (HGNC, https://www.genenames.org/) [498].

## Figures and Tables

**Figure 1 ijms-20-04381-f001:**
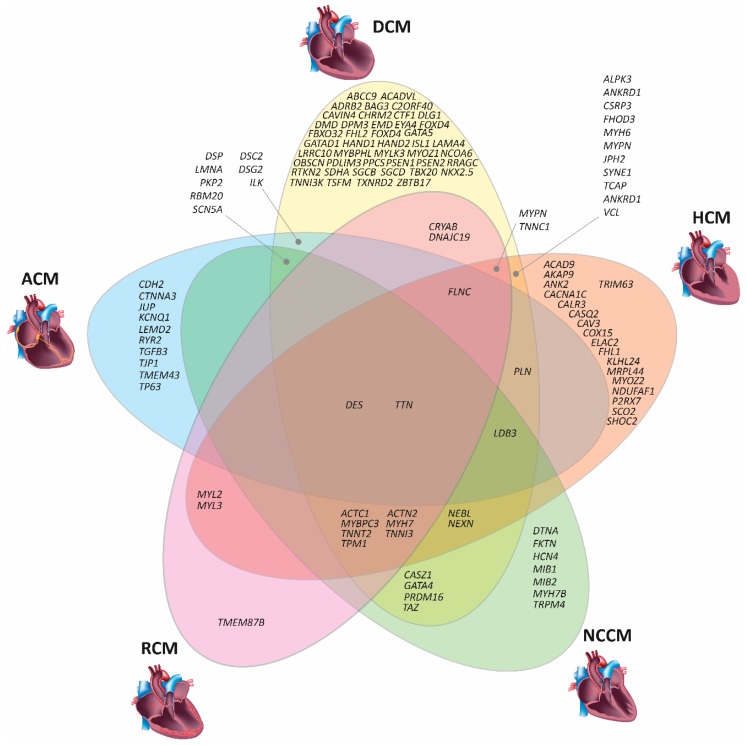
Schematic overview on cardiomyopathy associated genes and related clinical phenotypes. DCM—Dilated cardiomyopathy. HCM—Hypertrophic cardiomyopathy, ACM—Arrhythmogenic cardiomyopathy, NCCM—Non-compaction cardiomyopathy, RCM—Restrictive cardiomyopathy (Images of the DCM or HCM heart were licensed from shutterstock.com).

**Figure 2 ijms-20-04381-f002:**
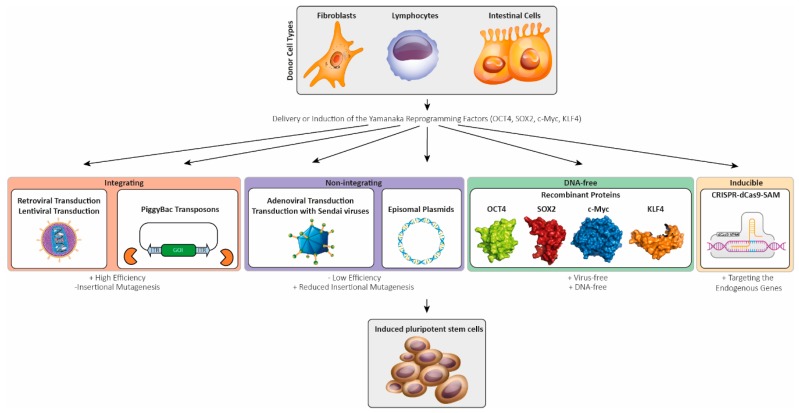
Schematic overview about different delivery methods of the Yamanaka factors into somatic primary cells for reprogramming (sub-figures for the cell types and viruses were licensed from shutterstock.com).

**Table 1 ijms-20-04381-t001:** Overview of cardiomyopathy associated genes carrying mutations.

Gene	Protein	Function	HCM	DCM	NCCM	ACM	RCM
*ABCC9*	ATP Binding Cassette Subfamily C Member 9	ABC transporter		[77]			
*ACAD9*	Acyl-CoA Dehydrogenase Member 9	Dehydrogenase	[78]				
*ACADVL*	Acyl-CoA Dehydrogenase Very Long Chain	Dehydrogenase		[79]			
*ACTC1*	Cardiac Actin	Sarcomere protein	[43,80]	[81]	[82]		[83]
*ACTN2*	α-Actinin 2	Z-band protein	[84]	[85]	[86]		[69]
*ADRB2*	Adrenoreceptor β2	G-protein coupled receptor		[87]			
*AKAP9*	A Kinase Anchoring Protein 9	Scaffolding protein	[88]				
*ALMS1*	Alstrom Syndrome Protein 1	Microtubule organization		[89] ^1^			
*ALPK3*	α-Kinase 3	Kinase	[90]	[90]			
*ANK2*	Ankyrin 2	Cytoskeleton linker protein	[91]			[92]	
*ANKRD1*	Ankyrin Repeat Domain Containing Protein 1	Transcription factor	[93]	[94,95]			
*BAG3*	Bcl-2 Associated Athanogene 3	Co-chaperone		[96]			[69,97]
*BRAF*	B-Raf Proto-Oncogene, Serine/Threonine Kinase	Kinase	[98] ^2^				
*C2ORF40*	Chromosome 2 Open Reading Frame 40	Hormone		[99]			
*CACNA1C*	Calcium Voltage-Gated Channel Subunit α1C	Calcium channel	[100]				
*CALM3*	Calmodulin 3	Calcium binding	[101] ^3^				
*CALR3*	Calreticulin 3	Calcium binding chaperone	[46]				
*CASQ2*	Calsequestrin 2	Calcium binding	[46]				
*CASZ1*	Castor Zinc Finger 1	Transcription factor		[102]	[103]		
*CAV3*	Caveolin 3	Scaffolding protein	[104]				
*CAVIN4*	Muscle Restricted Coiled Coil Protein	Myofibrillar organization		[105]			
*CDH2*	N-Cadherin	Cell–cell adhesion				[106,107]	
*CHRM2*	Cholinergic Receptor Muscarinic 2	G-protein coupled receptor		[108]			
*COL3A1*	Collagen Type III Alpha 1 Chain	Extra cellular matrix protein			[109] ^4^		
*COX15*	Cytochrome C Oxidase Assembly Homolog COX15	Mitochondrial respiratory chain	[110]				
*CRYAB*	αB-Crystallin	Chaperone-like activity		[111]			[112]
*CSRP3*	Muscle LIM Protein	Scaffolding protein	[113,114,115]	[116]			
*CTF1*	Cardiotrophin 1	Cytokine		[117]			
*CTNNA3*	αT-Catenin	Cell–cell adhesion				[118]	
*DES*	Desmin	Intermediate filament protein	[25]	[24,119]	[30]	[26]	[28]
*DLG1*	Discs Large MAGUK Scaffold Protein 1	Scaffolding protein		[88]			
*DMD*	Dystrophin	Dystrophin–glycoprotein complex		[120]			
*DNAJC19*	DNAJ Heat Shock Protein Family C19	Co-chaperone		[121]			[121]
*DOLK*	Dolichol Kinase	Phosphorylation of dolichol		[122] ^5^			
*DPM3*	Dolichyl-Phosphate Mannosyltransferase Subunit 3	Mannosyltransferase		[123]			
*DSC2*	Desmocollin 2	Cell–cell adhesion		[35]		[124]	
*DSG2*	Desmoglein 2	Cell–cell adhesion		[125]		[126,127]	
*DSP*	Desmoplakin	Cell–cell adhesion		[128]	[129]	[130]	
*DTNA*	α-Dystrobrevin	Dystrophin-glycoprotein complex			[131]		
*ELAC2*	ElaC Ribonuclease Z2	3′-tRNA endoribonuclease	[132]				
*EMD*	Emerin	Nuclear lamina associated protein		[133]			
*EYA4*	Eyes Absent Homolog 4	Transcription factor		[51]			
*FBN1*	Fibrillin 1	Extra cellular matrix protein	[134] ^6^	[135] ^7^	[136] ^7^		
*FBXO32*	F-Box Only Protein 32	Ubiquitin–protein ligase complex		[137,138]			
*FHL1*	Four and a Half LIM Domain Protein 1	Scaffolding protein	[41]				
*FHL2*	Four and a Half LIM Domain Protein 2	Scaffolding protein		[139]			
*FHOD3*	Formin Homology 2 Domain Containing Protein 3	Organization of actin-polymerization	[140]	[141]			
*FKRP*	Fukutin Related Protein	Posttranslational modification of dystroglycan		[142] ^8^			
*FKTN*	Fukutin	Glycosyltransferase of dystroglycan			[143]		
*FLNC*	Filamin C	Cell junction organization	[45]	[144,145]		[145]	[72]
*FOXD4*	Forkhead Box Protein D4	Transcription factor		[146]			
*FXN*	Frataxin	Regulation of mitochondrial iron transport	[147] ^9^				
*GAA*	α-Glucosidase	Glycogen metabolism	[148] ^9^				
*GATA4*	GATA Binding Protein 4	Transcription factor		[149]	[150] ^10^		
*GATA5*	GATA Binding Protein 5	Transcription factor		[151]			
*GATAD1*	GATA Zink Finger Domain Containing Protein 1	Gene expression regulation		[152]			
*GLA*	Galactosidase α	Galactose metabolism	[153] ^11^				
*GTPBP3*	GTP Binding Protein 3, Mitochondrial	Mitochondrial tRNA modification	[154] ^12^				
*HAND1*	Heart and Neural Crest Derivatives Expressed 1	Transcription factor		[155]			
*HAND2*	Heart and Neural Crest Derivatives Expressed 2	Transcription factor		[156]			
*HCN4*	Hyperpolarization Activated Cyclic Nucleotide Gated Potassium Channel 4	Potassium channel			[157]		
*HRAS*	HRas Proto-Oncogene GTPase	Signaling protein	[158]^13^				
*ILK*	Integrin Linked Kinase	Scaffolding protein		[159,160]		[68]	
*ISL1*	ISL LIM Homeobox 1	Transcription factor		[161]			
*ITGA7*	Integrin Subunit A7	Cell–cell and cell–matrix junction protein			[162] ^14^		
*ITPA*	Inosine Triphosphate Pyrophosphatase	Nucleotide metabolism		[163] ^15^			
*JPH2*	Junctophilin 2	Junctional complex	[164]	[165]			
*JUP*	Plakoglobin	Cell–cell adhesion				[58]	
*KCNQ1*	Potassium Channel Voltage Gated KQT-Like Subfamily Member 1	Potassium channel				[166]	
*KLHL24*	Kelch Like 24	Ubiquitin ligase substrate receptor	[167]				
*LAMA4*	Laminin α4	Extra cellular matrix protein		[159]			
*LAMP2*	Lysosomal Associated Membrane Protein 2	Chaperone-mediated autophagy	[168] ^16^				
*LDB3*	LIM Domain Binding Protein 3	Z-band protein	[169]	[170,171]	[170,172]	[173]	
*LEMD2*	LEM Domain Containing Protein 2	Nuclear lamina associated protein				[64,174] ^17^	
*LMNA*	Lamin A/C	Nuclear lamina associated protein		[49]	[175]	[63]	
*LRRC10*	Leucine Rich Repeat Containing Protein 10	Actin and α-actinin binding protein		[176]			
*MIB1*	Mindbomb Drosophila Homolog 1	Ubiquitin ligase			[177]		
*MIB2*	Mindbomb Drosophila Homolog 2	Ubiquitin ligase			[178] ^18^		
*MRPL3*	Mitochondrial Ribosomal Protein L3	Mitochondrial ribosomal protein	[179] ^19^				
*MRPL44*	Mitochondrial Ribosomal Protein L44	Mitochondrial ribosomal protein	[180,181]				
*MYBPC3*	Myosin Binding Protein C3	Sarcomere protein	[182,183]	[184]	[185]		[186]
*MYBPHL*	Myosin Binding Protein H-Like	Sarcomere protein		[187]			
*MYH6*	Myosin Heavy Chain 6	Sarcomere protein	[188]	[188]			
*MYH7*	Myosin Heavy Chain 7	Sarcomere protein	[22]	[48]	[7]		[189]
*MYH7B*	Myosin Heavy Chain 7B	Sarcomere protein			[162] ^20^		
*MYL2*	Myosin Light Chain 2	Sarcomere protein	[190]				[191]
*MYL3*	Myosin Light Chain 3	Sarcomere protein	[192]				[192]
*MYLK3*	Myosin Light Chain Kinase 3	Kinase		[193]			
*MYOZ1*	Myozenin 1	Calcineurin interacting protein		[194]			
*MYOZ2*	Myozenin 2	Calcineurin interacting protein	[195]				
*MYPN*	Myopalladin	Z-band protein	[196]	[94,197]			[196,198]
*NCOA6*	Nuclear Receptor Coactivator 6	Gene expression regulation		[199]			
*NDUFAF1*	NADH: Ubiquinone Oxidoreductase Complex Assembly Factor 1	Mitochondrial respiratory chain	[200]				
*NDUFV2*	NADH: Ubiquinone Oxidoreductase Core Subunit V2	Mitochondrial respiratory chain	[201,202] ^21^				
*NEBL*	Nebulette	Z-band protein	[203]	[204]	[203]		
*NEXN*	Nexilin	Sarcomere protein	[205]	[206]	[207]		
*NKX2.5*	NK2 Homeobox 5	Transcription factor		[208]			
*OBSCN*	Obscurin	Scaffolding protein		[209]			
*P2RX7*	Purinergic receptor P2X7	ATP gated ion channel	[210]				
*PDLIM3*	PDZ And LIM Domain 3	Z-band protein		[194]			
*PKP2*	Plakophilin 2	Cell-cell adhesion		[35]	[211]	[212]	
*PLN*	Phospholamban	Regulator of SERCA	[46]	[213,214]		[67]	
*PPCS*	Phosphopantothenoylcystein Synthetase	Co-enzyme A synthesis		[215]			
*PRDM16*	PR Domain Containing Protein 16	Transcription factor		[216]	[217]		
*PRKAG2*	Protein Kinase AMP Activated Non-catalytic G2	Energy sensor kinase	[218,219] ^22^				
*PSEN1*	Presenilin 1	γ-Secretase		[220,221]			
*PSEN2*	Presenilin 2	γ-Secretase		[220]			
*PTEN*	Phosphatase and Tensin Homolog	Phosphatase			[150] ^23^		
*PTPN11*	Protein Tyrosine Phosphatase Non-Receptor Type 1	Phosphatase	[222] ^24^				
*RAF1*	Raf-1 Proto-Oncogene, Serine/Threonine Kinase	Kinase	[223,224] ^25^	[225]			
*RBM20*	RNA Binding Protein 20	Splicing factor		[52,226]	[227]	[228,229]	
*RRAGC*	Ras Related GTP Binding C	GTR/RAG GTP-binding protein		[230]			
*RTKN2*	Rhotekin 2	Scaffolding protein		[99]			
*RYR2*	Ryanodine Receptor 2	Calcium channel				[66]	
*SCN5A*	Sodium Channel Voltage Gated Type V Subunit A	Sodium channel		[50,231]		[232]	
*SCO2*	SCO2 Cytochrome C Oxidase Assembly Protein	Metallo-chaperone	[233]				
*SDHA*	Succinate Dehydrogenase Complex Subunit A	Mitochondrial respiratory chain		[234]			
*SGCB*	Sarcoglycan β	Dystrophin-glycoprotein complex		[235]			
*SGCD*	Sarcoglycan δ	Dystrophin-glycoprotein complex		[236]			
*SHOC2*	Suppressor Of Clear, C. Elegans, Homolog	Scaffolding protein	[237]				
*SYNE1*	Nesprin 1	Component of the LINC complex	[238]	[239]			
*TAZ*	Tafazzin	Cardiolipin metabolism		[240] ^26^	[241,242]		
*TBX20*	T-Box Factor 20	Transcription factor		[243,244]			
*TCAP*	Thelethonin	Titin binding	[245]	[244,245]			
*TGFB3*	Transforming Growth Factor β3	Growth factor				[246]	
*TJP1*	Zonula Occludens 1	Tight junction adapter protein				[247]	
*TMEM43*	Transmembrane Protein 43	Nuclear lamina associated protein				[9,10]	
*TMEM87B*	Transmembrane Protein 87B	Endosome-to-trans-Golgi retrograde transport					[248]
*TNNC1*	Cardiac Troponin C	Sarcomere protein	[39]	[249]			[250]
*TNNI3*	Cardiac Troponin I	Sarcomere protein	[40]	[251]	[252]		[71]
*TNNI3K*	TNNI3 Interacting Kinase	Kinase		[253]			
*TNNT2*	Cardiac Troponin T	Sarcomere protein	[38]	[254]	[255]		[83]
*TP63*	Tumor Protein 63	Transcription factor				[256]	
*TPM1*	Tropomyosin 1	Sarcomere protein	[38,257]	[258]	[259]		[191]
*TRIM63*	Tripartite Motif Containing Protein 63	Ubiquitin ligase	[260]				
*TRPM4*	Transient Receptor Potential Cation Channel Subfamily M	Cation channel			[261]		
*TSFM*	Mitochondrial Translation Elongation Factor Ts	Translation elongation factor		[262]			
*TTN*	Titin	Sarcomere protein	[263]	[32,264]	[87,265]	[33]	[34]
*TTR*	Transthyretin	Carrier protein	[266,267] ^27^				
*TXNRD2*	Thioredoxin Reductase 2	Reduces thioredoxins		[268]			
*VCL*	Vinculin	Cell–cell and cell–matrix junction protein	[269,270]	[271]			
*ZBTB17*	Zinc Finger and BTB Domain Containing Protein 17	Transcription factor		[272,273]			

**^1^** Alström syndrome (MIM #203800); **^2^** Cardiofaciocutaneous syndrome (MIM #115150); **^3^** Modifier gene; **^4^** Ehlers–Danlos syndrome (MIM #130090); **^5^** Multi-organ involvement; **^6^** Digenetic with *PTPN11* mutations, combined with Marfan and Leopard syndrome; **^7^** Marfan Syndrome (MIM #154700); **^8^** Limb-girdle muscular dystrophy; **^9^** Friedreich ataxia (MIM #229300); **^10^** Digenetic with *PTEN*; ^11^ Fabry disease; **^12^** In combination with lactic acidosis and encephalopathy; **^13^** Costello syndrome (MIM #218040); **^14^** Digenetic with *MYH7B*;^**15**^ Martsolf-like syndrome (MIM #212720) in combination with DCM; **^16^** Danon disease (MIM #300257); **^17^** In combination with cataract; **^18^** In combination with giant hypertrophic gastritis (MIM #137280, Ménétrier disease); **^18^** In combination with psychomotor retardation; **^19^** Digenetic with *ITGA7;*
**^20^** In combination with encephalopathy; **^21^** Wolff–Parkinson–White syndrome (MIM #194200); **^22^** Digenetic with *GATA4* mutation; **^23^** Noonan syndrome; **^24^** Noonan syndrome or Leopard syndrome; **^25^** Barth syndrome (MIM #302060); **^26^** Amyloid cardiomyopathy (MIM #105210); ^27^ Fabry disease.

**Table 2 ijms-20-04381-t002:** Overview about important iPSC lines carrying mutations in genes associated with genetic cardiomyopathies or related diseases.

Gene	Protein	Mutation(s)	Method of Generation	Main Phenotypic Findings	Associated Disease	References
*ACTC1*	Cardiac Actin	p.E99K	▪ Sendai virus transduction ▪ Isogenic controls using CRISPR-Cas9 (PiggyBac)	Arrhythmias	HCM/LVNC	[380]
*ALPK3*	α-Kinase 3	p.W1264X^hom^	Electroporation with episomal plasmids	▪ Sarcomeric disarray ▪ Ca^2+^ handling defects	HCM	[381]
*BAG3*	Bcl-2 Associated Athanogene 3	▪ p.R90X ▪ p.R90X^hom^ ▪ p.R123X	▪ Electroporation with episomal plasmids ▪ & genome editing using ▪ CRISPR-Cas9 ▪ TALENs	▪ Decreased BAG3 expression ▪ Sarcomeric disarray after prolonged culture ▪ Decreased contraction	DCM	[374]
*BRAF*	B-Raf Proto-Oncogene, Serine/Threonine Kinase	▪ p.Q257R ▪ p.T599R	▪ Retroviral transduction ▪ Electroporation with episomal plasmids	▪ Cellular hypertrophy ▪ Pro-hypertrophic gene expression ▪ Ca^2+^ handling defects ▪ Abnormal TGFβ signaling	CFCS/HCM	[382]
*CAV*	Caveolin	▪ c.303G > C ▪ c.233C > A ▪ c.∆184-192	Electroporation with episomal plasmids	NA	MP	[383]
*CRYAB*	αB-Crystallin	▪ c.343delT^het^ ▪ c.343delT^hom^	Retroviral transduction and genome editing (zinc finger nucleases)	▪ No detectable expression of mutant αB-Crystallin ▪ Loss of function mechanism	MFM	[384]
*DES*	Desmin	p.N116S	Lentiviral transduction	NA	ACM	[385]
*DES*	Desmin	c.735+1G > A	Sendai virus transduction	NA	DRC	[386]
*DES*	Desmin	p.A285V	Retroviral transduction	▪ Desmin aggregation ▪ Z-disk streaming ▪ Decreased spontaneous beating rate	DCM	[387]
*DMD*	Dystrophin	▪ ∆Ex8-12 ▪ c.5899C > T	Sendai virus transduction	▪ Electrophysiological alterations ▪ Arrhythmias ▪ Prolonged action potential	DMD	[388]
*DMD*	Dystrophin	▪ ∆Ex8-9 ▪ ∆Ex6-9 ▪ ∆Ex7-11 ▪ ∆Ex3-9	Sendai virus transduction in combination with CRISPR-Cas9	▪ Out of frame deletion ∆Ex8-9 reduce contraction force ▪ Second deletions to correct the reading fame of DMD restores the contractility	DMD	[379]
*DMD*	Dystrophin	▪ c.263delG ▪ ∆Ex50	Lentiviral transduction CRISPR-Cas9	▪ Reduced contractility ▪ Ca^2+^ handling defects	DMD	[389,390]
*DSG2*	Desmoglein-2	p.G638R	Sendai virus transduction	▪ Electrophysiological alterations ▪ Ion channel dysfunction	ACM	[391]
*DSP*	Desmoplakin	p.R451G	Sendai virus transduction & genome editing for correction (CRISPR-Cas9)	Reduced desmoplakin expression	ACM	[392]
*FBN1*	Fibrillin 1	c.4028G > A	Sendai virus transduction	NA	Marfan Syndrome (HCM)	[393]
*FKRP*	Fukutin Related Protein	c.826C > A^hom^	Lentiviral transduction	▪ Abnormal action potential ▪ Electrophysiological alterations ▪ Decreased expression of *SCN5A* and *CACNA1C*	Limb-Girdle Muscular Dystrophy (DCM)	[394]
*FXN*	Frataxin	Expanded GAA repeats	Retroviral transduction	▪ Iron homeostasis defects ▪ Disorganized mitochondria ▪ Cellular hypertrophy ▪ Increased BNP expression ▪ Ca^2+^ handling defects	Friedreich Ataxia (HCM)	[395]
*FXN*	Frataxin	Expanded GAA repeats ▪ 800/600 ▪ 900/400	Lentiviral transduction	▪ Impaired mitochondrial function ▪ Decreased mitochondrial membrane potential ▪ Degeneration of mitochondria	Friedreich Ataxia (HCM)	[396]
*GLA*	Galactosidase α	IVS4+919G > A	Retroviral transduction	▪ Decreased α-galactosidase activity ▪ Cellular hypertrophy ▪ Upregulation of fibrotic genes	Fabry Disease (HCM)	[397,398]
*LAMP2*	Lysosomal Associated Membrane Protein 2	IVS6+1_4delGTGA	Sendai virus transduction	Autophagy dysfunction	Danon Disease (CM)	[399]
*LAMP2*	Lysosomal Associated Membrane Protein 2	▪ c.129-130insAT ▪ IVS-1.c64+1G > A	Unknown	▪ Mitochondrial-oxidative stress ▪ Apoptosis ▪ Disrupted mitophagic flux ▪ Mitochondrial respiratory deficiency	Danon Disease (CM)	[400]
*LAMP2*	Lysosomal Associated Membrane Protein 2	▪ c.1082delA ▪ c.247C > T ▪ c.64+1G > A	▪ Retroviral transduction ▪ Sendai virus transduction ▪ CRISPR-Cas9 for correction	▪ Defects in autophagic fusion ▪ Mitochondrial abnormalities ▪ Contractile abnormalities	Danon Disease (CM)	[401]
*LMNA*	Lamin A/C	p.S143P	Sendai virus transduction	▪ Sarcomere damage after hypoxia ▪ Arrhythmias after β-adrenergic stimulation ▪ Ca^2+^ handling defects	DCM	[402]
*LMNA*	Lamin A/C	p.S18fsX	Combined lentiviral and retroviral transduction	Normal nuclear membrane morphology	DCM	[403]
*LMNA*	Lamin A/C	p.R225X	Lentiviral transduction	▪ Reduced expression of lamin A/C ▪ Increased cellular apoptosis under electrical stimulation	DCM	[404]
*LMNA*	Lamin A/C	▪ p.R225X ▪ p.Q354X ▪ p.T518fsX29	Lentiviral transduction	▪ Increased nuclear blebbing under electrical stimulation ▪ Increased apoptosis under electrical stimulation ▪ Haploinsufficiency ▪ Treatment with PTC124 reverse the phenotypic findings	DCM & conduction disorders	[405]
*LMNA*	Lamin A/C	p.K219T	Lentiviral transduction	▪ Electrophysiological alterations ▪ Downregulation of *SCN5A* expression by epigenetic modulation of the promoter	DCM & conduction disorders	[406]
*MT-RNR2*	Mitochondrially Encoded 16S rRNA	m.2336T > C	Retroviral transduction	▪ Decreased stability of 16S rRNA ▪ Mitochondrial dysfunction ▪ Reduced ATP/ADP ratio ▪ Reduced mitochondrial potential ▪ Electrophysiological alterations	HCM	[407]
*MYBPC3*	Myosin Binding Protein C3	▪ p.V321M ▪ p.V219L ▪ c.2905+1G > A	Sendai virus transduction	Abnormal Ca^2+^ handling	HCM	[408]
*MYBPC3*	Myosin Binding Protein C3	p.R326Q	Electroporation with episomal plasmids	Ca^2+^ handling deficits	HCM	[409]
*MYBPC3*	Myosin Binding Protein C3	c.2373	Lentiviral transduction	▪ Cellular hypertrophy ▪ Contractile defect	HCM	[410,411]
*MYBPC3*	Myosin Binding Protein C3	p.R502W	Electroporation with episomal plasmids	NA	HCM	[412]
*MYBPC3*	Myosin Binding Protein C3	▪ p.R502W ▪ p.W792VfsX41	CRISPR-Cas9	▪ Hypercontractility ▪ P53 activation ▪ Oxidative stress ▪ Metabolic stress	HCM	[413]
*MYBPC3*	Myosin Binding Protein C3	▪ p.R943X ▪ p.R1073fsX4	Sendai virus transduction & genome editing for correction (CRISPR-Cas9)	▪ Reduced expression of *MYBPC3* at the mRNA level but not at the protein level ▪ Ca^2+^ handling defects ▪ Activation of nonsense-mediated mRNA decay	HCM	[414,415]
*MYBPC3*	Myosin Binding Protein C3	p.G999-Q1004del	Sendai virus transduction	▪ Cellular hypertrophy ▪ Myofibrillar disarray ▪ Reduced *MYBPC3* expression ▪ Increased ANP expression	HCM	[416]
*MYBPC3*	Myosin Binding Protein C3	p.Q1061X	▪ Sendai virus transduction ▪ Retroviral transduction	Arrhythmias	HCM	[417,418]
*MYBPC3*	Myosin Binding Protein C3	p.V454CfsX21	Retroviral transduction	▪ Haploinsufficiency (at the mRNA and protein level) ▪ Cellular hypertrophy ▪ Altered gene expression ▪ Efficient gene replacement using AAV9 reduce phenotypic findings	HCM	[419]
*MYBPC3*	Myosin Binding Protein C3	∆25 bp in intron 32 including the splicing branch point & p.D389V (same allele)	Sendai virus transduction	▪ Cellular hypertrophy ▪ Ca^2+^ handling deficits	HCM	[420]
*MYBPHL*	Myosin Binding Protein H-Like	p.R255X	Electroporation with episomal plasmids	Haploinsufficiency by nonsense mediated mRNA decay	DCM & conduction disorders	[187]
*MYH7*	Myosin Heavy Chain 7	p.R663H	Sendai virus transduction	Abnormal Ca^2+^ handling	HCM	[408]
*MYH7*	Myosin Heavy Chain 7	▪ p.R453C^het^ ▪ p.R453C^hom^	CRISPR-Cas9	▪ Cellular hypertrophy ▪ Sarcomeric disarray ▪ Increased expression of hypertrophy markers ▪ Ca^2+^ handling deficits	HCM	[421]
*MYH7*	Myosin Heavy Chain 7	▪ p.R403Q ▪ p.V606M	CRISPR-Cas9	▪ Hypercontractility ▪ P53 activation ▪ Oxidative stress ▪ Metabolic stress	HCM	[413]
*MYH7*	Myosin Heavy Chain 7	p.V698A	Electroporation with episomal plasmids	NA	HCM	[422]
*MYH7*	Myosin Heavy Chain 7	p.E848G	Electroporation with episomal plasmids	Reduced contractile function	HCM	[423,424]
*MYH7*	Myosin Heavy Chain 7	p.R403Q	Electroporation with episomal plasmids	NA	HCM	[425]
*MYH7*	Myosin Heavy Chain 7	p.R633H	Lentiviral transduction	▪ Ca^2+^ handling deficits ▪ Arrhythmias ▪ Cellular hypertrophy	HCM	[414,426]
*MYH7*	Myosin Heavy Chain 7	p.R442G	Retroviral transduction	▪ Disorganized sarcomeres ▪ Increased expression of genes involved in cell proliferation ▪ Electrophysiological alterations	HCM	[427]
*MYL2*	Myosin Light Chain 2	p.R58Q	Non-integrating mRNA/miRNA technology	▪ Cellular hypertrophy ▪ Myofibrillar disarray ▪ Irregular contraction ▪ Decreased Ca^2+^ transients	HCM	[428]
*MYL3*	Myosin Light Chain 3	▪ p.A57D^het^ ▪ p.A57D^hom^ ▪ p.A57G^het^	CRISPR-Cas9	▪ Asymptomatic ▪ Classification of benign GSVs	HCM	[375]
*PKP2*	Plakophilin-2	p.L614P	Retroviral transduction	▪ Reduced expression of plakophilin-2 ▪ Adipogenic phenotype	ACM	[429]
*PKP2*	Plakophilin-2	▪ c.2484C > T^hom^ ▪ c.2013delC	Retroviral transduction	▪ Lipogenesis ▪ Apoptosis ▪ Ca^2+^ handling deficits ▪ Pro-fibrotic gene expression ▪ Dysregulation of genes, encoding cell-cell connections.	ACM	[430,431,432]
*PKP2*	Plakophilin-2	c.972insT	Retroviral transduction	▪ Reduced expression of plakophilin-2 ▪ Changes of the desmosomal structure ▪ Lipid droplet accumulation	ACM	[433]
*PKP2*	Plakophilin-2	▪ c.354delT ▪ p.K859R	Sendai virus transduction	NA	ACM	[434]
*PKP2*	Plakophilin-2	c.2569_3018del50	Electroporation with episomal plasmids	NA	ACM	[435]
*PLN*	Phospholamban	p.R9C	CRISPR-Cas9	▪ Cellular hypertrophy ▪ Ca^2+^ handling deficits ▪ Increased expression of hypertrophic markers ▪ Altered metabolic state ▪ Changes of miRNA expression ▪ Increased expression of profibrotic genes	DCM	[414,436]
*PLN*	Phospholamban	p.R14del	Transfection with mRNAs& genome editing (TALENs) for mutation correction	▪ Ca^2+^ handling deficits ▪ Abnormal cytoplasmic localization of phospholamban ▪ Increased expression of hypertrophic markers ▪ Gene correction reverses the phenotypic findings	DCM	[437,438]
*PRGAG2*	Protein Kinase AMP-Activated Non-Catalytic Subunit Gamma 2	p.R302Q	Sendai virus transduction & genome editing for correction (CRISPR-Cas9)	▪ Arrhythmias ▪ Electrophysiological alterations ▪ Cellular hypertrophy ▪ Gene correction using CRISPR-Cas9 reverses the phenotypic findings	Wolff–Parkinson–White Syndrome (HCM)	[439]
*PRKAG2*	Protein Kinase AMP-Activated Non-Catalytic Subunit Gamma 2	p.N488I	Lentiviral transduction & genome editing for correction (TALEN)	▪ Activated AMPK remodeled metabolism ▪ Cellular hypertrophy	HCM	[440]
*PTPN11*	Protein Tyrosine Phosphatase Non-Receptor Type 11	p.T468M	Retroviral transduction	▪ Cellular hypertrophy ▪ Impaired sarcomere structure	LEOPARD Syndrome (HCM)	[360]
*PTPN11*	Protein Tyrosine Phosphatase Non-Receptor Type 11	p.Q510P	Sendai virus transduction	NA	LEOPARD Syndrome (HCM)	[441]
*RAF1*	Raf-1 Proto-Oncogene, Serine/Threonine Kinase	p.S257L	Electroporation of episomal plasmids & genome editing for correction (CRISPR-Cas9)	▪ Cellular hypertrophy ▪ Myofibrillar disarray ▪ Hyperactivation of MEK1/2 pathway ▪ Increased ERK5 signaling	Noonan Syndrome (HCM)	[442]
*RBM20*	RNA Binding Motif Protein 20	p.S635A	Lentiviral transduction	▪ Altered Ca^2+^ handling ▪ Impaired sarcomere structure ▪ Reduced titin N2B isoform expression	DCM	[443]
*RBM20*	RNA Binding Motif Protein 20	p.R636S	Sendai virus transduction	▪ Impaired sarcomere structure ▪ Altered transcriptome ▪ Altered Ca^2+^ handling ▪ Apoptotic changes ▪ Therapeutic treatment using β-blockers or Ca^2+^ channel blockers reverse phenotypic findings	DCM	[444,445]
*RYR2*	Ryanodine Receptor 2	p.F2483I	Retroviral transduction	▪ Arrhythmias ▪ Altered Ca^2+^ handling	CPVT	[350]
*RYR2*	Ryanodine Receptor 2	▪ p.S404R & p.N685S ▪ p.G3946S & p.G1885E	Sendai virus transduction	▪ Altered Ca^2+^ handling ▪ Calmodulin-dependent protein kinase II inhibition reverse the arrhythmias	CPVT	[376]
*SCN5A*	Sodium Voltage-Gated Channel Alpha Subunit 5	▪ p.S1898R	Sendai virus transduction & CRISPR-Cas9 for correction	▪ Reduction in peak sodium channel	ACM	[446]
*SCN5A*	Sodium Voltage-Gated Channel Alpha Subunit 5	p.R219H	Sendai virus transduction	▪ Proton leakage ▪ Disrupted ion homeostasis ▪ Structural abnormalities ▪ Electrophysiological alterations ▪ Reduced contraction	ACM/DCM	[447]
*SCO2*	SCO2 Cytochrome C Oxidase Assembly Protein	▪ p.E140K ▪ p.G193S^hom^	Sendai virus transduction	▪ Structural abnormalities ▪ Altered Ca^2+^ handling	HCM	[448]
*TAZ*	Tafazzin	▪ c.517delG ▪ c.328T > C	Transfection with synthetic mRNAs & CRISPR-Cas9 for correction	▪ Impaired sarcomere structure ▪ Decreased contraction ▪ Increased reactive oxygen species	Barth Syndrome	[449]
*TBX20*	T-Box Factor 20	▪ p.T262M ▪ p.Y317X	Sendai virus transduction	▪ Perturbed TGFβ signaling ▪ Reduced expression of cardiac transcription factors	LVNC	[450]
*TNNT2*	Cardiac Troponin T	p.R92W	Sendai virus transduction & CRISPR-Cas9 for correction	Abnormal Ca^2+^ handling	HCM	[408]
*TNNT2*	Cardiac Troponin T	p.R173W	Lentiviral transduction	▪ Decreased contractility ▪ Altered Ca^2+^ handling ▪ Impaired sarcomere structure	DCM	[414,451,452,453,454]
*TNNT2*	Cardiac Troponin T	▪ Compound heterozygous: ∆5bp and ∆2bp deletions in exon 2 leading to frameshifts ▪ Heterozygous ∆27bp deletion in exon 2 leading to a frameshift	TALEN	▪ Sarcomere disassembly ▪ Altered Ca^2+^ handling	DCM/HCM	[453]
*TNNT2*	Cardiac Troponin T	p.I79N	CRISPR-Cas9	▪ Impaired sarcomere structure ▪ Increased systolic function ▪ Impaired relaxation ▪ Altered Ca^2+^ handling	HCM	[455,456]
*TPM1*	Tropomyosin-1	p.D175N	▪ Sendai virus transduction ▪ Retroviral transduction	Arrhythmias	HCM	[417,418]
*TTN*	Titin	▪ p.W976R^+/-^ ▪ p.V6382fs^+/-^ ▪ p.V6382fs^-/-^ ▪ p.A22352fs^+/-^ ▪ p.P22582fs^+/-^ ▪ p.N22577fs^+/-^ ▪ p.N22577fs^-/-^ ▪ p.T33520fs^-/-^	▪ Lentiviral transduction (for patient specific iPSC) ▪ CRISPR-Cas9 (for generation of isogenic iPSC)	▪ Impaired sarcomere structure ▪ Decreased contractility ▪ Diminished activation of growth factors, hypoxia regulating factors and MAP kinases	DCM	[457]
*TTN*	Titin	p.S14450fsX4	Sendai virus transduction	Antisense-mediated exon skipping restores titin expression	DCM	[377]
*TTN*	Titin	▪ c.86076dupA ▪ c.70690dupAT	Lentiviral transduction	▪ Sarcomere defects ▪ Diminished inotropic and lusitropic responses	DCM	[458]
*TTR*	Transthyretin	p.L55P	Lentiviral transduction	Increased oxidative stress	Hereditary Transthyretin Amyloidosis	[459]

ACM—Arrhythmogenic cardiomyopathy; CFCS—Cardio facio cutaneous syndrome; CM—Cardiomyopathy; CPVT—Catecholaminergic polymorphic ventricular tachycardia; DCM—Dilated cardiomyopathy; DMD—Duchenne muscular dystrophy; DRC—Desmin-related cardiomyopathy; HCM—Hypertrophic cardiomyopathy; LVNC—Left-ventricular non-compaction cardiomyopathy; MFM—Myofibrillar myopathy; MP—Myopathy; NA—Not assessed; RCM—Restrictive cardiomyopathy.

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
