# Peer review of "Human Induced Pluripotent Stem-Cell-Derived Cardiomyocytes as Models for Genetic Cardiomyopathies"

_ijms, 2019, doi:10.3390/ijms20184381_

Round 1
Reviewer 1 Report
This review can be very relevant for research in cardiomyopathies but needs further improvement. Both, in terms of readability and content.
The main purpose of the review is difficult to grasp at this point. Authors should provide a clear idea of the potential usefulness of this review.
The criteria used for selecting what to include (or exclude) seems erratic at times. For example, the weight assigned to the history of iPS cells or the history of patch clamping confuses the reader about the main issues that are considered in the present review.
What mechanistics insights can be learned from Table 1?
Table 2 shows around 70 publications using hiPS-CM to study cardiomyopathies. A comprenhensive educated summary of that extensive literature will be greatly appreciated.
Authors should provide their own point view highlighting what can be learned from the literature cited. It will be helpful if authors can summarize what are the main answers already obtained from these models and the remaining fundamental questions in the field. Comparison and critical assessment is missing.
Author Response
We want to thank both reviewers for reviewing our manuscript. In the following, we will answer point-to-point to the reviewers.
Reviewer #1-1
The main purpose of the review is difficult to grasp at this point. Authors should provide a clear idea of the potential usefulness of this review.
Answer of the authors and changes of the manuscript
We thank reviewer #1 for this helpful comment. In the abstract of the revised version of the manuscript, we explained the usefulness of our review article in more detail.
Reviewer #1-2
The criteria used for selecting what to include (or exclude) seems erratic at times. For example, the weight assigned to the history of iPS cells or the history of patch clamping confuses the reader about the main issues that are considered in the present review.
Answer of the authors and changes of the manuscript
We think that the history of iPSC generation is relevant for our review article because significant improvements were achieved in the last decade(s), which are worth mentioning. However, we agree that the history of patch clamping is a little bit out of scope. According to the suggestion of reviewer #1, we have shortened therefore the revised manuscript at this point.
Reviewer #1-3
What mechanistics insights can be learned from Table 1?
Answer of the authors and changes of the manuscript
We thank reviewer #1 for this helpful comment. We inserted an additional column describing shortly the function of the mentioned genes/proteins. In addition, we cite Table 1 at relevant positions within the main text.
Reviewer #1-4
Table 2 shows around 70 publications using hiPS-CM to study cardiomyopathies. A compre[n]hensive educated summary of that extensive literature will be greatly appreciated.
Answer of the authors and changes of the manuscript
We thank reviewer #1 for this supportive comment. Therefore, we updated and extended the summary paragraph accordingly.
Reviewer #1-5
Authors should provide their own point view highlighting what can be learned from the literature cited. It will be helpful if authors can summarize what are the main answers already obtained from these models and the remaining fundamental questions in the field. Comparison and critical assessment is missing.
Answer of the authors and changes of the manuscript
As mentioned to point #1-4, we have inserted a summarizing paragraph to the different iPSC-CM models. In addition, we want to underline that a whole paragraph explains the limitations of iPSCs. Furthermore, we believe that the combination of iPSCs with animal models and /or explanted human myocardial tissue might be a key for therapy development in future. Therefore, we have updated the summary accordingly.
Reviewer #2-1
The authors provided an interesting overview about the genetic landscape of inherited cardiomyopathies, focusing on the development on human iPSC lines for modelling human cardiomyopathies in vitro and the current methods used for the cellular and molecular characterization of human iPSC-derived cardiomyocytes. I believe that the authors have provided a well-written and useful review on the emerging and sophisticated iPSC method used to model and investigate genetic cardiomyopathies. Each concept is reported comprehensively and exhaustively described, and is good represented by clear tables and figures. I believe that the review can be published in its present form.
Answer of the authors
We thank reviewer #2 for these motivating statement about our review article.
Reviewer 2 Report
The Authors provided an interesting overview about the genetic landscape of inherited cardiomyopathies, focusing on the development on human iPSC lines for modelling human cardiomyopathies in vitro and the current methods used for the cellular and molecular characterization of human iPSC-derived cardiomyocytes.
I believe that the Authors have provided a well-written and useful review on the emerging and sophisticated iPSC method used to model and investigate genetic cardiomyopathies.
Each concept is reported comprehensively and exhaustively described, and is good represented by clear tables and figures.
I believe that the review can be published in its present form.
Author Response
Reviewer #2-1
The authors provided an interesting overview about the genetic landscape of inherited cardiomyopathies, focusing on the development on human iPSC lines for modelling human cardiomyopathies in vitro and the current methods used for the cellular and molecular characterization of human iPSC-derived cardiomyocytes. I believe that the authors have provided a well-written and useful review on the emerging and sophisticated iPSC method used to model and investigate genetic cardiomyopathies. Each concept is reported comprehensively and exhaustively described, and is good represented by clear tables and figures. I believe that the review can be published in its present form.
Answer of the authors
We thank reviewer #2 for these motivating statement about our review article.
Round 2
Reviewer 1 Report
I consider the authors have addressed most of my concerns.